# Caudal Duplication Syndrome Systematic Review—A Need for Better Multidisciplinary Surgical Approach and Follow-Up

**DOI:** 10.3390/medicina56120650

**Published:** 2020-11-27

**Authors:** Spătaru Radu-Iulian, Avino Adelaida, Iozsa Dan-Alexandru, Ivanov Monica, Serban Dragos, Tomescu Luminiţa Florentina, Cirstoveanu Cătălin

**Affiliations:** 1Discipline of Pediatric Surgery, Faculty of Medicine, “Carol Davila” University of Medicine and Pharmacy, 020021 Bucharest, Romania; radu_spataru@yahoo.com (S.R.-I.); dan.iozsa@yahoo.com (I.D.-A.); 2Department of Pediatric Surgery, Emergency Clinic Hospital for Children “Maria S. Curie”, 41451 Bucharest, Romania; mqmivanov@yahoo.com; 3Department of Plastic and Reconstructive Surgery, Clinical Emergency Hospital “Prof. Dr. Agrippa Ionescu”, 011356 Bucharest, Romania; 4Discipline of Plastic and Reconstructive Surgery, Faculty of Medicine, Doctoral School, “Carol Davila” University of Medicine and Pharmacy, 020021 Bucharest, Romania; 5Discipline of General Surgery, Faculty of Medicine, “Carol Davila” University of Medicine and Pharmacy, 020021 Bucharest, Romania; dragos.serban@umfcd.ro; 6Department of General Surgery, Emergency University Hospital, 050098 Bucharest, Romania; 7Department of Interventional Radiology, Clinical Emergency Hospital “Prof. Dr. Agrippa Ionescu”, 011356 Bucharest, Romania; slumi2001@gmail.com; 8Discipline of Pediatrics, Faculty of Medicine, “Carol Davila” University of Medicine and Pharmacy, 020021 Bucharest, Romania; cirstoveanu@yahoo.com; 9Neonatal Intensive Care Unit, Emergency Clinic Hospital for Children “Maria S. Curie”, 41451 Bucharest, Romania

**Keywords:** caudal duplication syndrome, colorectal duplication, genitourinary duplication, congenital malformation, pediatric surgery

## Abstract

*Background and Objectives:* Caudal duplication syndrome is a rare association of anatomical anomalies describing duplication of the hindgut, spine, and uro-genital structures, leading to varied clinical presentations. The current literature focuses on case reports which describe the embryological etiology and anatomical spectrum of the condition giving little attention to the surgical preparation, the need for a well-structured follow-up program, or the transition into adult healthcare of these complex patients. No reviews have been published regarding this complex pathology. *Materials and Methods*: A review of caudal duplication syndrome cases was done to assess the range of the clinical malformations, timing, and types of surgical interventions. Inconsistencies in multidisciplinary care, follow-up, and risk events were described. *Results:* Hindgut duplication always involved the anorectal region. Anorectal malformations were evenly distributed as unilateral and bilateral. Colon duplication extended from the anal region to the transverse colon or ascending colon in most of the cases and less to terminal. In females, genital duplication was present in all cases. The follow-up period varied between 3 months and 12 years. In all adult females, the motive of presentation was related to pregnancy (complications after successful delivery, fertility evaluation) or late complications (fecalith obstruction of the end-to-side colon anastomosis, repeated UTIs with renal scarring). *Conclusions*: Complex malformations affecting multiple caudal organs may have a strong impact in many aspects of the long-term quality of life; therefore, patients with caudal duplication syndrome need increased awareness and joined multidisciplinary treatment.

## 1. Introduction

Caudal duplication syndrome (CDS) is a very rare congenital cluster of anomalies. Dominguez et al. (1993) defined it as an association of hindgut duplication, duplication of the lower uro-genital tract, spinal cord, and vertebral anomalies at heterogeneous degrees of severity resulting from a fetal insult at different stages of embryogenesis [1,2]. 

Considering the very low incidence of this malformation and the variety of clinical presentations, the majority of published literature focus on description of the anatomical spectrum and the possible embryological etiology and less on the short- and long-term surgical planning, implementation of a structured follow-up plan, and the responsibility for transition of care into adult healthcare of these complex patients [3,4].

Each type of gastrointestinal, genitourinary, and/or distal spine duplication, either partial or complete cannot be discussed separately but together and its management must be adjusted to each individual case. The purpose of treatment is to preserve or improve fecal and urinary continence, maintain reproductive potential, allow a satisfactory sexual life with acceptable cosmetic appearance of the perineum, and manage other neurological disabilities. All these will impact the long-term morbidities and quality of life [5].

The impact of the malformations severity on the prognosis and risks of potential complications of the surgical treatment can be reduced by a comprehensive anatomical and functional preoperative evaluation, prospective multidisciplinary surgical planning, and active involvement of the patient and family to assure collaboration and adherence to the follow-up plan. 

The objective of this review is to recognize possible surgical management patterns used to treat different clinical presentations of CDS, to examine the presence of gaps in patients’ care, to assess risk events which can occur during life and might signal the need for planned intervention and identify groups of patients with CDS which require special attention.

## 2. Materials and Methods

A literature review with the search term “caudal duplication syndrome” was done using open access search engines PubMed, Science Direct, and Google Scholar by two independent reviewers. We excluded duplicate references, conference abstracts, articles not in English, and cases which did not describe the surgical management. Articles which described patients with duplicated digestive system, genitourinary tract, spinal column, and the neural tube but also included duplications of the lower limb were not included. A total of 279 articles were retrieved during the systematic search using the specified criteria. The full text articles were reviewed, and the selected articles were saved using reference management software. A total of 17 articles (3 case series and 14 case reports) with a total of 23 patients were selected for meeting the criteria. No literature or systematic review articles could be found.

## 3. Results

Hindgut duplication in CDS always included the anorectal region (Table 1).

At anal level, anorectal malformation (ARM) was present at least on one duplicated side in all cases with the exception of three patients without hindgut duplication and one patient with intra-sphincteric location of both anal openings. Unilateral anorectal malformation was present in 7/23 cases (30.43%), bilateral ARM in 7/23 cases (30.43%), and five cases were not described. The most common types of anorectal malformation are perineal fistula or recto vestibular fistula. Colon duplication extended from anal region to the transverse colon in 5/23 cases (21.73%), to ascending colon in 4/23 cases (17.39%), and to terminal ileum in 2/23 (8.69%). In eight patients, the level of duplication was not specified. In one patient, the appendix and proximal colon was duplicated while the sigmoid and anorectal region was triplicated. Bladder and urethral duplication were always in the sagittal plane and was present in all cases with the exception of two female patients (one with unspecified anatomy). In males, genital involvement with complete or partial shaft duplication (glans duplication and one shaft) was present in five out of seven patients. In females, genital duplication was present in all cases. Spinal cord malformations (myelomeningocele, tethered cord, cord lipoma, hydrosyrix) were reported in half of the cases. The vertebral spine, most commonly defects of fusion and hemivertebrae, was involved in 14/23 cases (60.86%). Associated anomalies outside of the caudal region included the abdominal wall (omphalocele), cardiac malformations (patent ductus arteriosus, ventricular septal defect, atrial septal defect), gastro-intestinal (Meckel diverticulum, malrotation, duodenal atresia, small bowel atresia, esophageal duplication cyst), or limb malformations (unilateral lower leg hypoplasia), without being consistent in prevalence. Prenatal ultrasound evaluation was done in only 2 patients without a prenatal diagnosis of CDS and no further Magnetic resonance imaging (MRI) evaluation. In 19/23 cases (82.60%), the child was evaluated in the first year of life due to the evident malformations of the perineum. In cases with non-obstructive symptoms due to less severe malformations or inconsistent follow-up, evaluation was delayed until complications occurred. The most common surgical interventions in the first month of life were colostomy for anorectal malformation with obstructive symptoms, acute abdomen (entero-vesical fistula) or omphalocele repair. After this age, the motive of presentation was related to complications: severe dermatitis because continuous dribbling urine, fecal incontinence, long term constipation or for cosmetic correction of the perineal region in asymptomatic female patients. Functional evaluation of the bladder is inconsistently reported and was done in cases of urinary incontinence. The most common associated urologic pathology was unilateral or bilateral vesico-ureteral reflux. In male patients, penectomy or penile reconstruction was done in 3/5 cases or was planned for future reconstruction in the rest of the cases. Only one author opted for vaginoplasty with resection of the common wall for cosmetic reasons in two patients and in one other case one side of the duplication was removed. All adult cases (age between 22 and 39 years) were females and the motive of presentation was related to pregnancy (complications after successful delivery, fertility evaluation) or late complications (fecalith obstruction of the end-to-side colon anastomosis, repeated UTIs with renal scarring). All cases had limited surgical history (colorectal surgery in infancy) with no subsequent events. All four female patients were sexually active with one or bilateral side vaginal use retained the double vagina, all became pregnant and delivered by cesarean section. The main reasons for delayed presentation were non-obstructive colorectal malformation (stooling present with the help with suppository), no need for toilet training until school age (incases with urinary or fecal incontinence), or lack of caregiver awareness. The follow-up period varied between 3 months and 12 years and was longer in cases with fecal or urinary incontinence. Three of the adult females with history of colorectal surgery as infants had no reported follow-up until presentation for current problems as adults. The life events with the highest impact on occurrence of complications were the type of colorectal surgical procedure (end-to-side anastomosis with recurrent episodes of fecal impaction which required multiple hospital admissions and treatment), neglect or lack of awareness from caregivers (patients presented at 6 and 13 years old, respectively, with neurologic and continence problems) and pregnancy. A case of a 39-year-old female is presented with a history of three pregnancies, delivered with cesarean section and no prior medical history. After deliveries, the patient had one side hysterectomy for leiomyomas, two interventions for vaginal prolapse, one side ureteral reimplantation, three transurethral incisions of the bladder neck for inefficient bladder emptying, and one transurethral bulking agent injection in the bladder neck for stress incontinence.

## 4. Discussion

The very low incidence of caudal duplication syndrome makes it difficult to develop expertise and assemble a multidisciplinary team with participation in all the steps of care. Because of the evident clinical features of the perineum, these cases are first assessed at birth. Prenatal examination can diagnose clusters of severe anomalies which might not be recognized as CDS because of the low awareness of this syndrome [6,7].


**Management of colorectal duplication**


If the baby has efficient bowel movements, is preferably breastfed, is gaining weight, and no other severe associated malformations require surgical intervention, we recommend delaying further treatment until a complete assessment of the types and extension of malformations is done and an individually tailored treatment plan can be made bringing the family together with the multidisciplinary team. The type of malformation determines the surgical strategy at birth: primary repair (not the best choice in these complex patients), temporary anal calibrations to ensure efficient bowel movements until definitive repair or colostomy creation. The purpose of surgical reconstruction is to have a single anal opening located within the complex muscle either by removal of the anal side opening outside of the sphincter [7,8] or joining of the two anorectal ends and positioning in the muscle sphincter [9]. Depending on the duplication length and anatomical variant, if the two duplicated colons are not fused and have a separate blood supply, one side should be removed if not, stripping of the mucosa on the non-dominant side [10] or stapling the common wall [9] (similar with stapling the common wall in Duhamel procedure [11]) to ensure efficient emptying, to avoid complications such as severe constipation [12] with fecal impaction, volvulus, or neoplastic changes. End-to-side anastomosis of the two colonic lumens should not be the first option because of the risk of stenosis at the anastomotic site in the long-term with fecal impaction and proximal colitis [3]. This can also make difficult efficient emptying of both lumens if antegrade continence will be required in patients with low the potential for fecal continence [9]. 


**Management of bladder and urethral duplication**


Sagittal urethral duplication (side-by-side) is associated with bladder duplication and is specific for CDS. The two bladders can be completely separate or have a sagittal septum (most commonly two asymmetric sides) each with a ureter from the ipsilateral kidney and its own urethra. Urologic management in the first year of life is almost always conservative. Anatomic evaluation should be complemented with bladder function assessment especially in symptomatic patients. The goal is to avoid recurrent urinary infections, to ensure there is efficient emptying of both bladders and urinary continence. In most female patients, unilateral removal of the urethra is not necessary if asymptomatic. In males, the urethra will be excised when the decision for cosmetical penectomy is made [13], and it can be postponed until adolescence to let the patient decide, to assess penile growth and erectile function.

Assessment of continence potential is important in the development of any anorectal and urinary reconstruction plan. The type and severity of spinal cord and sacrum malformation have a direct role in predicting the fecal and urinary continence [14,15]. This can be assessed very early on and will influence the choice of surgical procedures as for example the decision to remove the bladder septum creating a bigger capacity bladder with bladder neck ligation and ipsilateral urethral removal or to keep the one or two appendices for antegrade continent enema (ACE)and/or appendicovesicostomy [16].


**Management of genital duplication**


Gender distribution shows a predilection of CDS in females with a complete but well-developed duplicated external and internal genital organ and preserved ovarian function. Almost always, corrective genital surgery in females is not necessary unless one side is hypoplastic and might cause menstrual flow obstruction or for cosmetic concerns, in which case, it can be removed [10]. If congenital uterine anomalies such as septate or subseptate uteri have a risk for reduced conception rate, increased risk of first trimester miscarriage (especially with septal implantation), preterm birth, and fetal malpresentation, patients with didelphys uterus do not appear to have reduced fertility and have less pregnancy complications, but might be at increased risk for preterm labor [17]. All our adult female patients carried pregnancies to term and in three out the four cases [3,18,19,20] presented for pregnancy related issues (genetic counseling, symptomatic ureteral compression or vaginal prolapse, and voiding dysfunction after multiple pregnancies). Cesarean section was performed in all cases. In cases with a history of multiple abdominal surgeries and depending on the complexity of procedures, the cesarean surgery should be assisted by a colorectal surgeon and/or urologist. Male patients present with complete or partial penile duplication and are more likely to be corrected for cosmetic reasons with removal of one of the abnormal shafts [13] or penoplasty [12]. There are no reports assessing fertility and sexual function into adult age in CDS male patients. 


**Support groups and quality of life**


The importance of active involvement of family members and the patient itself is well recognized, but not always available. Parents must cope with the difficulty of accepting and adjusting to their child’s condition, coordinate appointments with different healthcare providers, find information about their child’s malformations, manage the financial demands of long-term medical care, and locate appropriate care centers [21,22]. Factors such as low income, socially disadvantaged groups, and inaccessible educational materials due to rarity of the malformations, limited access to information or lack of support groups can negatively influence adherence to the treatment plan. Quality of life in patients with CDS is unreported but it is known the long-term impact of incontinence and sexual dysfunction in patients with anorectal malformations. Spinal malformations such as myelomeningocele can lead to lower limb paralysis. The goal of management plan is to reduce disability early on by implementing bowel management programs in cases with low continence potential and the need for robust transitional care arrangements to enable continued management in adulthood [19,23]. The process of transition should start in early adolescence and the adult care provider who will assume care of the patient should be identified early in the process, when possible [3].

Our literature review has numerous limitations which influence the value and objectivity in making recommendations. We are aware of the difficulty of extracting consistent data and providing statistically significant results. There is a high heterogeneity in data reporting in case reports or case series articles, with ambiguous nomenclature used to describe the anatomy or details regarding surgical procedures and incomplete information about the extent of multi-organ malformations. The term “caudal duplication syndrome” was introduced in 1993, but there still are inconsistent ways of naming it (e.g., cloacal duplication).

## 5. Conclusions

Caudal duplication syndrome should receive more attention from the pediatric surgical community regarding team approach and data collection. It is not sufficient to report the anatomical particularities of a case. Similar to other rare and complex malformations which affect the caudal region and have an impact on the quality of life (for example, bladder exstrophy or cloacal exstrophy), patients with caudal duplication syndrome deserve a better integrated care to improve outcomes.

## Figures and Tables

**Table 1 medicina-56-00650-t001:** Spectrum of malformations in patients with caudal duplication syndrome (CDS).

Author Name	Patient ID	Age at Presentation (Gender)	Colorectal Duplication Extension (Proximal Level)	Type of Anal Duplication	Urinary Duplication	Genital Duplication	Spinal Malformation	Follow-up Period
Dominguez et al.	1	Newborn (M)	No duplication	No duplication	Complete	Diphallia, scrotum duplication	Meningo -myelocele, duplication of L5 and sacrum	7 years
	2	Newborn (F)	Transverse colon	Unilateral ARM	Complete	Complete	Myelomeningocele, Chiari-Arnold syndrome, scoliosis, dislocated hip, equinovarus leg	9 years
	3	Newborn (F)	Transverse colon	ARM (not specified)	Complete	Complete	T10 hemivertebra	2 years
	4	Newborn (M)	Double appendix, double proximal colon, sigmoid triplication	Triple ani (not specified)	Complete	No duplication, 4 testicles	Hemivertebrae, sacrum duplication	No Follow-up
Acer et al.	5	Newborn (F)	Appendix, cecum duplication	Bilateral ARM	Complete	Complete	Bifid L5, sacrum	3 months
Swaika et al.	6	Newborn (M)	Ascending colon	Bilateral ARM	Complete	2 Hemi-phalluses	Spinal lipoma, normal vertebral spine	1 year
Samuk et al.	7	Newborn (M)	Hepatic flexure	Bilateral ARM	Complete	No Duplication	Tethered cord, lipomyelo-meningocele, splitting bellow dorso-lumbar junction	No Follow-up
Bajpai et al.	8	Newborn (M)	Transverse colon	ARM (unspecified)	Complete	Diphallia	Lipomeningo-myelocele, tethered cord, bifid sacrum, pelvis diastasis	No Follow-up
Chaussy et al.	9	Newborn (F)	Terminal ileum	Unilateral ARM	Complete	Complete	Hemivertebrae	5 years
Kroes et al.	10	Newborn (F)	Colon (not specified)	Bilateral ARM	No duplication	Complete	Myelocele, duplication of lumbar and sacrum	1.5 years
	11	Newborn (F)	Terminal ileum	Bilateral ARM	Complete	Complete	Hemivertebrae, sacrum malformation	12 years
Wisenbaugh et al.	12	Newborn (F)	Colon (unspecified)	Bilateral ARM	Complete	Complete	Myelomeningocele tethered cord	7 years
de Oliveira et al.	13	9 months (M)	Colon (not specified)	ARM (unspecified)	Complete	Diphallia	Intramedullary L4 cyst, tethered cord, lipoma, sacrum duplication	No Follow-up
Bansal et al.	14	2 years 10 months (F)	Ascending colon	Unilateral ARM	Complete	Complete	Lipomyelo-meningocele, tethered cord, hydrosyrinx, scoliosis, butterfly vertebrae, hemi-sacrum	8 months
Abdelhalim et al.	15	6 months (F)	Distal colon (unspecified)	Bilateral ARM	Complete	Complete	No malformation	13 years
	16	4 years (F)	Colon (unspecified)	Parasagittal duplication with intra sphincteric location	Complete	Complete	Hemivertebrae, lumbar scoliosis	15 months
	17	6 years (F)	No duplication	No duplication	Complete	Complete	No duplication	1 year
Salman et al.	18	8 years (F)	Ascending colon	Bilateral ARM	Complete	Complete	Spina bifida, hemivertebrae	Not specified
Liu et al.	19	13 years (M)	Transverse colon	Unilateral ARM	Complete	Glans duplication One shaft	Hemivertebrae, sacrum subfissure	2 years
Becker et al.	20	22 years (F)	Colon (not specified)	Unilateral ARM	Unspecified	Complete	Segmentation defects, lumbar and sacrum fusion defects	Not specified
Hu et al.	21	28 years (F)	Colon (not specified)	Unilateral ARM	Complete	Complete	Lumbar and sacrum fusion defects	Not specified
Ragab et al.	22	31 years (F)	Colon (not specified)	ARM Not specified	Complete	Complete	N/A	No Follow-up
Mei et al.	23	39 years (F)	N/A	N/A	Complete	Complete	N/A	Not specified

F—female, M—male; ARM—anorectal malformation; Complete—duplication of bladder and urethra or utero-vaginal duplication.

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
