# Peer review of "Caudal Duplication Syndrome Systematic Review—A Need for Better Multidisciplinary Surgical Approach and Follow-Up"

_medicina, 2020, doi:10.3390/medicina56120650_

Round 1
Reviewer 1 Report
Interesting work, showing pediatric/ multidisciplinary approach to CDS.
Table 1- important, however column titles: spinal malformation and vertebral spine malformation are misleading. Perhaps it would be better: mail spinal malformation and other spinal defects?
Discussion - authors mentioned quality of life (QOL) and many factors influencing QOL, but one factor is missing: associated with CDS myelomeningocoele leads to lower limb paralysis. Many patients with CDS walk with crutches or use a wheelchair.
Author Response
Thank you for your remarks. We corrected them in the manuscript.
Reviewer 2 Report
The idea of the review, is very original and interesting. However, before publication I recommend some changes.
Title: In title authors suggest that they will consider multidisciplinary approach, however, in material and methods, and also in other sections they mainly consider surgical approach. Maybe title should be changer for more suitable to the presented data.
Introduction: The 1st reference does not support the statement in those lines.
Materials and methods: In mentioned cases authors describe Caudal Duplication Syndrome as presence of duplication of the urinary tract or/and reproductive system or/ and distal digestive system or/ and vertebral spine and spinal malformation, that is why searching only this one phrase (Caudal Duplication Syndrome) limits the research. Furthermore, I enclose a few possible cases that can be also use in this review, despite that this cases does not include “Caudal Duplication Syndrome”
Cohen, N., Ahmed, M. N., Goldfischer, R., & Zaghloul, N. (2019). Persistent cloaca and caudal duplication in a monovular twin, a rare case report. International Journal of Surgery Case Reports, 60, 137–140. doi:10.1016/j.ijscr.2019.06.013
Nepple, K. G., Cooper, C. S., & Austin, J. C. (2009). Rare Variant of Bladder Exstrophy Associated With Urethral, Bladder, and Colonic Duplication. Urology, 73(4), 928.e1–928.e3. doi:10.1016/j.urology.2008.06.037
Results: Should be changed with accordance to materials and methods.
Abbreviations from lines 102,106,122 should be explained, if they are used for the first time in the text.
Discussion: I recommend to change “stool” for “defecate” or other more medical word.
The whole paper is quite challenging to read, and I wonder whether it will benefit from perhaps adding some bullet headings under discussion section to summarize your statements.
A lot of concepts are covered, however the manner is somewhat haphazard, which makes for challenging reading and stunts the clarity of thought for the reader.
I also could not find references to lines 146- 152.
Information in lines 196-198 should be removed, it is not specific only for treating this syndrome.
Author Response
Thank you for the remarks.
Answer 1: By multidisciplinary approach we mean different surgical specialties: general pediatric surgeon, colorectal pediatric surgeon, pediatric urologist, pediatric neurosurgeon, pediatric gynecologist and also transition of care into adulthood of children with complex congenital malformations with the matching adult physicians. This a rather new concept, in the process of implementation, in only a few centers in the world. Most of the case reports were managed by a single surgeon doing all the necessary surgical interventions. According to your suggestion we changed the title from “multidisciplinary approach” to “multidisciplinary Surgical approach”.
Introduction -
We removed both the statement and 1st reference.
Materials and Method:
We used the term caudal duplication syndrome in accordance with Dominguez (Reference 3) the one who tried to clarify the ambiguity related to this term. He does not include twin malformations. “Cloacal duplication” is another term used to describe caudal duplication syndrome. Many case reports don’t provide all the details to fully define the patients malformations as CDS or not. Because of this we chose to include only the cases named in accordance.
Cohen:
We specifically mentioned in” Materials and Methods”: “Articles which described patients with duplicated digestive system, genitourinary tract, spinal column and the neural tube but included also duplications of the lower limb were not included.” In Cohen’s case, “This is a monochorinonic-diamniotic twin born at 36 weeks with apgars of 9/9. She had a duplicated labia with two clitorises, and a partially formed accessory foot with 2 toes protruding from the right gluteal region.”
In this article the author describes the case as” The OEIScomplex” (omphalocele, exstrophy, imperforate anus, spinal defects) as it was defined by Carey (Carey JC, Greenbaum B, Hall BD. The OEIS complex (omphalocele,exstrophy, imperforate anus, spinal defects). Birth Defects OrigArtic Ser. 1978;14:253-263.),not as caudal duplication syndrome.
Thus, regarding your suggestions, we didn’t make the changes, trying to stick to the definition made by Domingues. It’s true that there are many reported cases with a single duplicated caudal organ (urethra, vulvae, anus) which might suggest an incomplete caudal duplication sdr, and, also, it is tempting to consider caudal duplication syndrome and heteropagus twins as a continuum spectrum, but the idea of the article is to emphasize the importance of multidisciplinary surgical management in patients with caudal duplication sdr only (as described by Domingues).
Abbreviations from lines 102,106,122 should be explained, if they are used for the first time in the text.
Was corrected.
Discussion: I recommend to change “stool” for “defecate” or other more medical word.
Was corrected.
The whole paper is quite challenging to read, and I wonder whether it will benefit from perhaps adding some bullet headings under discussion section to summarize your statements.
Was corrected.
A lot of concepts are covered, however the manner is somewhat haphazard, which makes for challenging reading and stunts the clarity of thought for the reader.
We rephrased several sentences, for better flow of reading.
I also could not find references to lines 146- 152.
It is our recommendation, as we mentioned in the article.
Information in lines 196-198 should be removed, it is not specific only for treating this syndrome.
We removed it.
We hope everything is ok now.
All the best from Bucharest,
Adelaida Avino
Round 2
Reviewer 2 Report
I think the authors revised the manuscript successfully. I don't have any requests to revise the article.